# Faster Ridge Regression via the Subsampled Randomized Hadamard Transform

**Yichao Lu**[1]    **Paramveer S. Dhillon**[2]    **Dean Foster**[1]    **Lyle Ungar**[2]
[1]Statistics (Wharton School), [2]Computer & Information Science
University of Pennsylvania, Philadelphia, PA, U.S.A
{dhillon|ungar}@cis.upenn.edu
foster@wharton.upenn.edu, yichaolu@sas.upenn.edu

## Abstract

We propose a fast algorithm for ridge regression when the number of features is much larger than the number of observations ($p \gg n$). The standard way to solve ridge regression in this setting works in the dual space and gives a running time of $O(n^2 p)$. Our algorithm *Subsampled Randomized Hadamard Transform- Dual Ridge Regression* (SRHT-DRR) runs in time $O(np \log(n))$ and works by preconditioning the design matrix by a Randomized Walsh-Hadamard Transform with a subsequent subsampling of features. We provide risk bounds for our SRHT-DRR algorithm in the fixed design setting and show experimental results on synthetic and real datasets.

## 1   Introduction

Ridge Regression, which penalizes the $\ell_2$ norm of the weight vector and shrinks it towards zero, is the most widely used penalized regression method. It is of particular interest in the $p > n$ case ($p$ is the number of features and $n$ is the number of observations), as the standard ordinary least squares regression (OLS) breaks in this setting. This setting is even more relevant in today's age of 'Big Data', where it is common to have $p \gg n$. Thus efficient algorithms to solve ridge regression are highly desirable.

The current method of choice for efficiently solving RR is [19], which works in the dual space and has a running time of $O(n^2 p)$, which can be slow for huge $p$. As the runtime suggests, the bottleneck is the computation of $\mathbf{X}\mathbf{X}^\top$ where $\mathbf{X}$ is the design matrix. An obvious way to speed up the algorithm is to subsample the columns of $\mathbf{X}$. For example, suppose $\mathbf{X}$ has rank $k$, if we randomly subsample $p_{subs}$ of the $p$ ($k < p_{subs} \ll p$) features, then the matrix multiplication can be performed in $O(n^2 p_{subs})$ time, which is very fast! However, this speed-up comes with a big caveat. If all the signal in the problem were to be carried in just one of the $p$ features, and if we missed this feature while sampling, we would miss all the signal.

A parallel and recently popular line of research for solving large scale regression involves using some kind of random projections, for instance, transforming the data with a randomized Hadamard transform [1] or Fourier transform and then uniformly sampling observations from the resulting transformed matrix and estimating OLS on this smaller data set. The intuition behind this approach is that these frequency domain transformations uniformlize the data and smear the signal across all the observations so that there are no longer any high leverage points whose omission could unduly influence the parameter estimates. Hence, a uniform sampling in this transformed space suffices. This approach can also be viewed as preconditioning the design matrix with a carefully constructed data-independent random matrix. This transformation followed by subsampling has been used in a variety of variations, including Subsampled Randomized Hadamard Transform (SRHT) [4, 6] and Subsampled Randomized Fourier Transform (SRFT) [22, 17].

In this paper, we build on the above line of research and provide a fast algorithm for ridge regression (RR) which applies a Randomized Hadamard transform to the columns of the $\mathbf{X}$ matrix and then samples $p_{subs} = O(n)$ columns. This allows the bottleneck matrix multiplication in the dual RR to be computed in $O(np\log(n))$ time, so we call our algorithm *Subsampled Randomized Hadamard Transform-Dual Ridge Regression* (SRHT-DRR).

In addition to being computationally efficient, we also prove that in the fixed design setting SRHT-DRR only increases the risk by a factor of $(1 + C\sqrt{\frac{k}{p_{subs}}})$ (where $k$ is the rank of the data matrix) w.r.t. the true RR solution.

## 1.1 Related Work

Using randomized algorithms to handle large matrices is an active area of research, and has been used in a variety of setups. Most of these algorithms involve a step that randomly projects the original large matrix down to lower dimensions [9, 16, 8]. [14] uses a matrix of i.i.d Gaussian elements to construct a preconditioner for least square which makes the problem well conditioned. However, computing a random projection is still expensive as it requires multiplying a huge data matrix by another random dense matrix. [18] introduced the idea of using structured random projection for making matrix multiplication substantially faster.

Recently, several randomized algorithms have been developed for kernel approximation. [3] provided a fast method for low rank kernel approximation by randomly selecting $q$ samples to construct a rank $q$ approximation of the original kernel matrix. Their approximation can reduce the cost to $O(nq^2)$. [15] introduced a random sampling scheme to approximate symmetric kernels and [12] accelerates [15] by applying Hadamard Walsh transform. Although our paper and these papers can all be understood from a kernel approximation point of view, we are working in the $p \gg n \gg 1$ case while they focus on large $n$.

Also, it is worth distinguishing our setup from standard kernel learning. Kernel methods enable the learning models to take into account a much richer feature space than the original space and at the same time compute the inner product in these high dimensional space efficiently. In our $p \gg n \gg 1$ setup, we already have a rich enough feature space and it suffices to consider the linear kernel $\mathbf{X}\mathbf{X}^\top$ [1]. Therefore, in this paper we propose a randomized scheme to reduce the dimension of $\mathbf{X}$ and accelerate the computation of $\mathbf{X}\mathbf{X}^\top$.

## 2 Faster Ridge Regression via SRHT

In this section we firstly review the traditional solution of solving RR in the dual and it's computational cost. Then we introduce our algorithm SRHT-DRR for faster estimation of RR.

### 2.1 Ridge Regression

Let $\mathbf{X}$ be the $n \times p$ design matrix containing $n$ i.i.d. samples from the $p$ dimensional independent variable (a.k.a. "covariates" or "predictors") $X$ such that $p \gg n$. $Y$ is the real valued $n \times 1$ response vector which contains $n$ corresponding values of the dependent variable $Y$. $\epsilon$ is the $n \times 1$ homoskedastic noise vector with common variance $\sigma^2$. Let $\hat{\beta}_\lambda$ be the solution of the RR problem, i.e.

$$\hat{\beta}_\lambda = \arg \min_{\beta \in p \times 1} \frac{1}{n}\|Y - \mathbf{X}\beta\|^2 + \lambda\|\beta\|^2 \tag{1}$$

The solution to Equation (1) is $\hat{\beta}_\lambda = (\mathbf{X}^\top\mathbf{X} + n\lambda\mathbf{I}_p)^{-1}\mathbf{X}^\top Y$. The step that dominates the computational cost is the matrix inversion which takes $O(p^3)$ flops and will be extremely slow when $p \gg n \gg 1$. A straight forward improvement to this is to solve the Equation (1) in the dual space. By change of variables $\beta = \mathbf{X}^\top\alpha$ where $\alpha \in n \times 1$ and further letting $\mathbf{K} = \mathbf{X}\mathbf{X}^\top$ the optimization problem becomes

$$\hat{\alpha}_\lambda = \arg \min_{\alpha \in n \times 1} \frac{1}{n}\|Y - \mathbf{K}\alpha\|^2 + \lambda\alpha^\top\mathbf{K}\alpha \tag{2}$$

and the solution is $\hat{\alpha}_\lambda = (\mathbf{K} + n\lambda I_n)^{-1}Y$ which directly gives $\hat{\beta}_\lambda = \mathbf{X}^\top \hat{\alpha}_\lambda$. Please see [19] for a detailed derivation of this dual solution. In the $p \gg n$ case the step that dominates computational cost in the dual solution is computing the linear kernel matrix $\mathbf{K} = \mathbf{X}\mathbf{X}^\top$ which takes $O(n^2p)$ flops. This is regarded as the computational cost of the true RR solution in our setup.

Since our algorithm SRHT-DRR uses Subsampled Randomized Hadamard Transform (SRHT), some introduction to SRHT is warranted.

## 2.2 Definition and Properties of SRHT

Following [20], for $p = 2^q$ where $q$ is any positive integer, a SRHT can be defined as a $p_{subs} \times p$ ($p > p_{subs}$) matrix of the form:

$$\Theta = \sqrt{\frac{p}{p_{subs}}}\mathbf{RHD}$$

where

- $\mathbf{R}$ is a random $p_{subs} \times p$ matrix the rows of which are $p_{subs}$ uniform samples (without replacement) from the standard basis of $\mathbb{R}^p$.

- $\mathbf{H} \in \mathbb{R}^{p \times p}$ is a normalized Walsh-Hadamard matrix. The Walsh-Hadamard matrix of size $p \times p$ is defined recursively: $H_p = \begin{bmatrix} H_{p/2} & H_{p/2} \\ H_{p/2} & -H_{p/2} \end{bmatrix}$ with $H_2 = \begin{bmatrix} +1 & +1 \\ +1 & -1 \end{bmatrix}$. $H = \frac{1}{\sqrt{p}}H_p$ is a rescaled version of $H_p$.

- $\mathbf{D}$ is a $p \times p$ diagonal matrix and the diagonal elements are i.i.d. Rademacher random variables.

There are two key features that makes SRHT a nice candidate for accelerating RR when $p \gg n$. Firstly, due to the recursive structure of the $\mathbf{H}$ matrix, it takes only $O(p \log(p_{subs}))$ FLOPS to compute $\Theta v$ where $v$ is a generic $p \times 1$ dense vector while for arbitrary unstructured $p_{subs} \times p$ dense matrix $\mathbf{A}$, the cost for computing $\mathbf{A}v$ is $O(p_{subs}p)$ flops. Secondly, after projecting any matrix $\mathbf{W} \in p \times k$ with orthonormal columns down to low dimensions with SRHT, the columns of $\Theta\mathbf{W} \in p_{subs} \times k$ are still about orthonormal. The following lemma characterizes this property:

**Lemma 1.** *Let $\mathbf{W}$ be an $p \times k$ ($p > k$) matrix where $\mathbf{W}^\top\mathbf{W} = \mathbf{I}_k$. Let $\Theta$ be a $p_{subs} \times p$ SRHT matrix where $p > p_{subs} > k$. Then with probability at least $1 - (\delta + \frac{p}{e^k})$,*

$$\|(\Theta\mathbf{W})^\top \Theta\mathbf{W} - \mathbf{I}_k\|_2 \leq \sqrt{\frac{c\log(\frac{2k}{\delta})k}{p_{subs}}} \tag{3}$$

The bound is in terms of the spectral norm of the matrix. The proof of this lemma is in the Appendix. The tools for the random matrix theory part of the proof come from [20] and [21]. [10] also provided similar results.

## 2.3 The Algorithm

Our fast algorithm for SRHT-DRR is described below:

---

**SRHT-DRR**
Input: Dataset $\mathbf{X} \in n \times p$, response $Y \in n \times 1$, and subsampling size $p_{subs}$.
Output: The weight parameter $\beta \in p_{subs} \times 1$.

- Compute the SRHT of the data: $\mathbf{X}_H = \mathbf{X}\Theta^\top$.
- Compute $\mathbf{K}_H = \mathbf{X}_H\mathbf{X}_H^\top$
- Compute $\alpha_{H,\lambda} = (\mathbf{K}_H + n\lambda\mathbf{I}_n)^{-1}Y$ , which is the solution of Equation (2) obtained by replacing $\mathbf{K}$ with $\mathbf{K}_H$.
- Compute $\beta_{H,\lambda} = \mathbf{X}_H^\top\alpha_{H,\lambda}$

---

Since, SRHT is only defined for $p = 2^q$ for any integer $q$, so, if the dimension $p$ is not a power of 2, we can concatenate a block of zero matrix to the feature matrix $\mathbf{X}$ to make the dimension a power of 2.

**Remark 1.** *Let's look at the computational cost of SRHT-DRR. Computing $\mathbf{X}_H$ takes $O(np \log(p_{subs}))$ FLOPS [2, 6]. Once we have $\mathbf{X}_H$, computing $\alpha_{H,\lambda}$ costs $O(n^2 p_{subs})$ FLOPS, with the dominating step being computing $\mathbf{K}_H = \mathbf{X}_H \mathbf{X}_H^\top$. So the computational cost for computing $\alpha_{H,\lambda}$ is $O(np \log(p_{subs}) + n^2 p_{subs})$, compared to the true RR which costs $O(n^2 p)$. We will discuss how large $p_{subs}$ should be later after stating the main theorem.*

# 3 Theory

In this section we bound the risk of SRHT-DRR and compare it with the risk of the true dual ridge estimator in fixed design setting.

As earlier, let $\mathbf{X}$ be an arbitrary $n \times p$ design matrix such that $p \gg n$. Also, we have $Y = \mathbf{X}\beta + \epsilon$, where $\epsilon$ is the $n \times 1$ homoskedastic noise vector with common mean 0 and variance $\sigma^2$. [5] and [3] did similar analysis for the risk of RR under similar fixed design setups.

Firstly, we provide a corollary to Lemma 1 which will be helpful in the subsequent theory.

**Corollary 1.** *Let $k$ be the rank of $\mathbf{X}$. With probability at least $1 - (\delta + \frac{p}{e^k})$*

$$(1 - \Delta)\mathbf{K} \preceq \mathbf{K}_H \preceq (1 + \Delta)\mathbf{K} \tag{4}$$

*where $\Delta = C\sqrt{\frac{k \log(2k/\delta)}{p_{subs}}}$. ( as for p.s.d. matrices $\mathbf{G} \succeq \mathbf{L}$ means $\mathbf{G} - \mathbf{L}$ is p.s.d.)*

*Proof.* Let $\mathbf{X} = \mathbf{U}\mathbf{D}\mathbf{V}^\top$ be the SVD of $\mathbf{X}$ where $\mathbf{U} \in n \times k$, $\mathbf{V} \in p \times k$ has orthonormal columns and $\mathbf{D} \in k \times k$ is diagonal. Then $\mathbf{K}_H = \mathbf{U}\mathbf{D}(\mathbf{V}^\top \Theta\Theta\mathbf{V})\mathbf{D}\mathbf{U}^\top$. Lemma 1 directly implies $\mathbf{I}_k(1 - \Delta) \preceq (\mathbf{V}^\top \Theta\Theta\mathbf{V}) \preceq \mathbf{I}_k(1 + \Delta)$ with probability at least $1 - (\delta + \frac{p}{e^k})$. Left multiply $\mathbf{U}\mathbf{D}$ and right multiply $\mathbf{D}\mathbf{U}^\top$ to the above inequality complete the proof. $\square$

## 3.1 Risk Function for Ridge Regression

Let $Z = \mathbb{E}_\epsilon(Y) = \mathbf{X}\beta$. The risk for any prediction $\hat{Y} \in n \times 1$ is $\frac{1}{n}\mathbb{E}_\epsilon \|\hat{Y} - Z\|^2$.
For any $n \times n$ positive symmetric definite matrix $\mathbf{M}$, define the following risk function.

$$R(\mathbf{M}) = \frac{\sigma^2}{n}\text{Tr}[\mathbf{M}^2(\mathbf{M} + n\lambda\mathbf{I}_n)^{-2}] + n\lambda^2 Z^\top(\mathbf{M} + n\lambda\mathbf{I}_n)^{-2}Z \tag{5}$$

**Lemma 2.** *Under the fixed design setting, the risk for the true RR solution is $R(K)$ and the risk for SRHT-DRR is $R(K_H)$.*

*Proof.* The risk of the SRHT-DRR estimator is

$$
\begin{aligned}
\frac{1}{n}\mathbb{E}_\epsilon\|\mathbf{K}_H\alpha_{H,\lambda} - Z\|^2 &= \frac{1}{n}\mathbb{E}_\epsilon\|\mathbf{K}_H(\mathbf{K}_H + n\lambda\mathbf{I}_n)^{-1}Y - Z\|^2 \\
&= \frac{1}{n}\mathbb{E}_\epsilon\|\mathbf{K}_H(\mathbf{K}_H + n\lambda\mathbf{I}_n)^{-1}Y - \mathbb{E}_\epsilon(\mathbf{K}_H(\mathbf{K}_H + n\lambda I_n)^{-1}Y)\|^2 \\
&\quad + \frac{1}{n}\|\mathbb{E}_\epsilon(\mathbf{K}_H(\mathbf{K}_H + n\lambda\mathbf{I}_n)^{-1}Y) - Z\|^2 \\
&= \frac{1}{n}\mathbb{E}_\epsilon\|\mathbf{K}_H(\mathbf{K}_H + n\lambda\mathbf{I}_n)^{-1}\epsilon\|^2 \\
&\quad + \frac{1}{n}\|(\mathbf{K}_H(\mathbf{K}_H + n\lambda\mathbf{I}_n)^{-1}Z - Z\|^2 \\
&= \frac{1}{n}\text{Tr}[\mathbf{K}_H^2(\mathbf{K}_H + n\lambda\mathbf{I}_n)^{-2}\epsilon\epsilon^\top] \\
&\quad + \frac{1}{n}Z^\top(\mathbf{I}_n - \mathbf{K}_H(\mathbf{K}_H + n\lambda\mathbf{I}_n)^{-1})^2 Z \\
&= \frac{\sigma^2}{n}\text{Tr}[\mathbf{K}_H^2(\mathbf{K}_H + n\lambda\mathbf{I}_n)^{-2}] \\
&\quad + n\lambda^2 Z^\top(\mathbf{K}_H + n\lambda\mathbf{I}_n)^{-2}Z
\end{aligned}
\tag{6}
$$

Note that the expectation here is only over the random noise $\epsilon$ and it is conditional on the Randomized Hadamard Transform. The calculation is the same for the ordinary estimator. In the risk function, the first term is the variance and the second term is the bias. □

## 3.2 Risk Inflation Bound

The following theorem bounds the risk inflation of SRHT-DRR compared with the true RR solution.

**Theorem 1.** *Let $k$ be the rank of the $\mathbf{X}$ matrix. With probability at least $1 - (\delta + \frac{p}{e^k})$*

$$
R(\mathbf{K}_H) \le (1 - \Delta)^{-2} R(\mathbf{K})
\tag{7}
$$

*where $\Delta = C\sqrt{\frac{k\log(2k/\delta)}{p_{subs}}}$*

*Proof.* Define

$$
\begin{aligned}
B(\mathbf{M}) &= n\lambda^2 Z^\top(\mathbf{M} + n\lambda\mathbf{I}_n)^{-2}Z \\
V(\mathbf{M}) &= \frac{\sigma^2}{n}\text{Tr}[\mathbf{K}_H^2(\mathbf{K}_H + n\lambda\mathbf{I}_n)^{-2}]
\end{aligned}
$$

for any p.s.d matrix $\mathbf{M} \in n \times n$. Therefore, $R(\mathbf{M}) = V(\mathbf{M}) + B(\mathbf{M})$. Now, due to [3] we know that $B(\mathbf{M})$ is non-increasing in $\mathbf{M}$ and $V(\mathbf{M})$ is non-decreasing in $\mathbf{M}$. When Equation(4) holds,

$$
\begin{aligned}
R(\mathbf{K}_H) &= V(\mathbf{K}_H) + B(\mathbf{K}_H) \\
&\le V((1+\Delta)\mathbf{K}) + B((1-\Delta)\mathbf{K}) \\
&\le (1+\Delta)^2 V(\mathbf{K}) + (1-\Delta)^{-2}B(\mathbf{K}) \\
&\le (1-\Delta)^{-2}(V(\mathbf{K}) + B(\mathbf{K})) \\
&= (1-\Delta)^{-2}R(\mathbf{K})
\end{aligned}
$$

□

**Remark 2.** *Theorem 1 gives us an idea of how large $p_{subs}$ should be. Assuming $\Delta$ (the risk inflation ratio) is fixed, we get $p_{subs} = C\frac{k\log(2k/\delta)}{\Delta^2} = O(k)$. If we further assume that $\mathbf{X}$ is full rank, i.e. $k = n$, then, it suffices to choose $p_{subs} = O(n)$. Combining this with Remark 1, we can see that the cost of computing $\mathbf{X}_H$ is $O(np\log(n))$. Hence, under the ideal setup where $p$ is huge so that the dominating step of SRHT-DRR is computing $\mathbf{X}_H$, the computational cost of SRHT-DRR $O(np\log(n))$ FLOPS.*

**Comparison with PCA**    Another way to handle high dimensional features is to use PCA and run regression only on the top few principal components (this procedure is called PCR), as illustrated by [13] and many other papers. RR falls in the family of "shrinkage" estimators as it shrinks the weight parameter towards zero. On the other hand, PCA is a "keep-or-kill" estimator as it kills components with smaller eigenvalues. Recently, [5] have shown that the risk of PCR and RR are related and that the risk of PCR is bounded by four times the risk of RR. However, we believe that both PCR and RR are parallel approaches and one can be better than the other depending on the structure of the problem, so it is hard to compare SRHT-DRR with PCR theoretically.

Moreover, PCA under our $p \gg n \gg 1$ setup is itself a non-trivial problem both statistically and computationally. Firstly, in the $p \gg n$ case we do not have enough samples to estimate the huge $p \times p$ covariance matrix. Therefore the eigenvectors of the sample covariance matrix obtained by PCA maybe very different from the truth. (See [11] for a theoretical study on the consistency of the principal directions for the high $p$ low $n$ case.) Secondly, PCA requires one to compute an SVD of the $\mathbf{X}$ matrix, which is extremely slow when $p \gg n \gg 1$. An alternative is to use a randomized algorithm such as [16] or [9] to compute PCA. Again, whether randomized PCA is better than our SRHT-DRR algorithm depends on the problem. With that in mind, we compare SRHT-DRR against standard as well as Randomized PCA in our experiments section; We find that SRHT-DRR beats both of them in speed as well as accuracy.

# 4    Experiments

In this section we show experimental results on synthetic as well as real-world data highlighting the merits of SRHT, namely, lower computational cost compared to the true Ridge Regression (RR) solution, without any significant loss of accuracy. We also compare our approach against "standard" PCA as well as randomized PCA [16].

In all our experiments, we choose the regularization constant $\lambda$ via cross-validation on the training set. As far as PCA algorithms are concerned, we implemented standard PCA using the built in SVD function in MATLAB and for randomized PCA we used the block power iteration like approach proposed by [16]. We always achieved convergence in three power iterations of randomized PCA.

## 4.1    Measures of Performance

Since we know the true $\beta$ which generated the synthetic data, we report MSE/Risk for the fixed design setting (they are equivalent for squared loss) as measure of accuracy. It is computed as $\|\hat{Y} - X\beta\|^2$, where $\hat{Y}$ is the prediction corresponding to different methods being compared. For real-world data we report the classification error on the test set.

In order to compare the computational cost of SHRT-DRR with true RR, we need to estimate the number of FLOPS used by them. As reported by other papers, e.g. [4, 6], the theoretical cost of applying Randomized Hadamard Transform is $O(np \log(p_{subs}))$. However, the MATLAB implementation we used took about $np \log(p)$ FLOPS to compute $\mathbf{X}_H$. So, for SRHT-DRR, the total computational cost is $np \log(p)$ for getting $\mathbf{X}_H$ and a further $2n^2 p_{subs}$ FLOPS to compute $\mathbf{K}_H$. As mentioned earlier, the true dual RR solution takes $\approx 2n^2 p$. So, in our experiments, we report relative computational cost which is computed as the ratio of the two.

$$\text{Relative Computational Cost} = \frac{p \log(p) \cdot n + 2n^2 p_{subs}}{2n^2 p}$$

## 4.2    Synthetic Data

We generated synthetic data with $p = 8192$ and varied the number of observations $n = 20, 100, 200$. We generated a $n \times n$ matrix $\mathbf{R} \sim MVN(\mathbf{0}, \mathbf{I})$ where $\text{MVN}(\mu, \mathbf{\Sigma})$ is the Multivariate Normal Distribution with mean vector $\mu$, variance-covariance matrix $\mathbf{\Sigma}$ and $\beta_j \sim \mathcal{N}(0, 1) \quad \forall j = 1, \ldots, p$. The final $\mathbf{X}$ matrix was generated by rotating $\mathbf{R}$ with a randomly generated $n \times p$ rotation matrix. Finally, we generated the Ys as $Y = \mathbf{X}\beta + \epsilon$ where $\epsilon_i \sim \mathcal{N}(0, 1) \quad \forall i = 1, \ldots, n$.

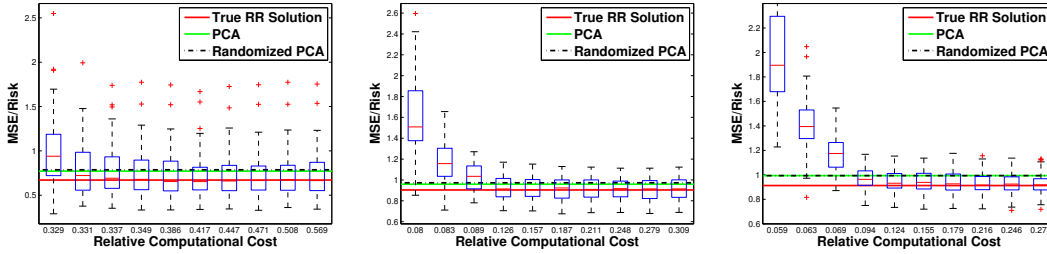

Figure 1: Left to right n=20, 100, 200. The boxplots show the median error rates for SRHT-DRR for different $p_{subs}$. The solid red line is the median error rate for the true RR using all the features. The green line is the median error rate for PCR when PCA is computed by SVD in MATLAB. The black dashed line is median error rate for PCR when PCA is computed by randomized PCA.

For PCA and randomized PCA, we tried keeping $r$ PCs in the range $10$ to $n$ and finally chose the value of $r$ which gave the minimum error on the training set. We tried $10$ different values for $p_{subs}$ from $n + 10$ to $2000$ . All the results were averaged over $50$ random trials.

The results are shown in Figure 1. There are two main things worth noticing. Firstly, in all the cases, SRHT-DRR gets very close in accuracy to the true RR with only $\approx 30\%$ of its computational cost. SRHT-DRR also cost much fewer FLOPS than the Randomized PCA for our experiments. Secondly, as we mentioned earlier, RR and PCA are parallel approaches. Either one might be better than the other depending on the structure of the problem. As can be seen, for our data, RR approaches are always better than PCA based approaches. We hypothesize that PCA might perform better relative to RR for larger $n$.

### 4.3 Real world Data

We took the UCI ARCENE dataset which has 200 samples with 10000 features as our real world dataset. ARCENE is a binary classification dataset which consists of 88 cancer individuals and 112 healthy individuals (see [7] for more details about this dataset). We split the dataset into 100 training and 100 testing samples and repeated this procedure 50 times (so $n = 100$, $p = 10000$ for this dataset). For PCA and randomized PCA, we tried keeping $r = 10, 20, 30, 40, 50, 60, 70, 80, 90$ PCs and finally chose the value of $r$ which gave the minimum error on the training set ($r = 30$). As earlier, we tried 10 different values for $p_{subs}$: $150, 250, 400, 600, 800, 1000, 1200, 1600, 2000, 2500$.

Standard PCA is known to be slow for this size datasets, so the comparison with it is just for accuracy. Randomized PCA is fast but less accurate than standard ("true") PCA; its computational cost for $r = 30$ can be approximately calculated as about $240np$ (see [9] for details), which in this case is roughly the same as computing $\mathbf{X}\mathbf{X}^{\top}$ ($\approx 2n^2p$).

The results are shown in Figure 2. As can be seen, SRHT-DRR comes very close in accuracy to the true RR solution with just $\approx 30\%$ of its computational cost. SRHT-DRR beats PCA and Randomized PCA even more comprehensively, achieving the same or better accuracy at just $\approx 18\%$ of their computational cost.

## 5 Conclusion

In this paper we proposed a fast algorithm, *SRHT-DRR*, for ridge regression in the $p \gg n \gg 1$ setting SRHT-DRR preconditions the design matrix by a Randomized Walsh-Hadamard Transform with a subsequent subsampling of features. In addition to being significantly faster than the true dual ridge regression solution, SRHT-DRR only inflates the risk w.r.t. the true solution by a small amount. Experiments on both synthetic and real data show that SRHT-DRR gives significant speeds up with only small loss of accuracy. We believe similar techniques can be developed for other statistical methods such as logistic regression.

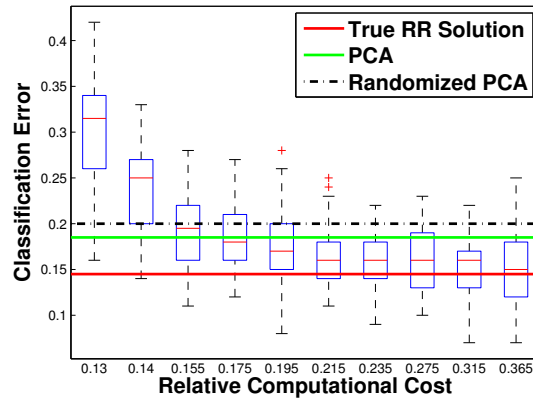

Figure 2: The boxplots show the median error rates for SRHT-DRR for different $p_{subs}$. The solid red line is the median error rate for the true RR using all the features. The green line is the median error rate for PCR with top 30 PCs when PCA is computed by SVD in MATLAB. The black dashed line is the median error rate for PCR with the top 30 PCs computed by randomized PCA.

## Footnotes

[1]For this reason, it is standard in natural language processing applications to just use linear kernels.

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
