[Supplementary Material]

# Supplementary Material for 'Faster Ridge Regression via the Subsampled Randomized Hadamard Transform'

**Yichao Lu**[1]    **Paramveer S. Dhillon**[2]    **Dean Foster**[1]    **Lyle Ungar**[2]
[1]Statistics (Wharton School), [2]Computer & Information Science
University of Pennsylvania, Philadelphia, PA, U.S.A
{dhillon|ungar}@cis.upenn.edu
foster@wharton.upenn.edu, yichaolu@sas.upenn.edu

Firstly, we would like to state some lemmas and give some properties of Subsampled Randomized Hadamard Transform (SRHT), which will be pivotal in proving our theorems for the fixed design setting.

## 1   Properties of SRHT

As described in the paper, let $\mathbf{H}$ be the scaled Hadamard matrix of size $p \times p$, $\mathbf{D}$ be the diagonal matrix of size $p \times p$ with i.i.d. rademacher random variable on the diagonal and let $\mathbf{R} \in p_{subs} \times p$ be the subsampling matrix. So, $\Theta = \mathbf{RHD} \in p_{subs} \times p$ is the SRHT matrix. All the norms used in this paper and supplementary material are $\ell_2$ norms for a vector and the spectral norm for a matrix unless specified otherwise. The statement of the lemma is as follows:

**Lemma 1.** *Let $\mathbf{X}$ be an $n \times p$ $(n \gg p)$ matrix where $\mathbf{X}^\top \mathbf{X} = n \cdot \mathbf{I}_p$. Let $\Theta$ be a $n_{subs} \times n$ SRHT matrix where $n_{subs}$ is the subsampling size. Then with failure probability at most $\delta + \frac{n}{e^p}$,*

$$\|(\Theta \mathbf{X})^\top \Theta \mathbf{X}/n_{subs} - \mathbf{X}^\top \mathbf{X}/n\| \leq \sqrt{\frac{c \log(\frac{2p}{\delta})p}{n_{subs}}} \tag{1}$$

**Remark 1.** *The idea and tools for the proof of this lemma come from [1] and [2]. Here we characterize the spectral norm error between the matrix multiplication with and without SRHT as a function of subsample size $n_{subs}$ and matrix dimension $p$.*

Before proving Lemma 1 we need to state a few lemmas from random matrix theory. Next Lemma is Lemma 3.3 in [1].

**Lemma 2.** *(Row norms after Randomized Hadamard Transform) Let $\mathbf{V}$ be an $n \times p$ matrix with orthonormal columns. Then $\mathbf{HDV}$ is also an $n \times p$ matrix with orthonormal columns and*

$$\mathbf{P}\left(\max_{j=1,2\ldots n} \|e_j^\top (\mathbf{HDV})\| \geq \sqrt{\frac{p}{n}} + \sqrt{\frac{8\log(\beta n)}{n}}\right) \leq \frac{1}{\beta} \tag{2}$$

**Remark 2.** *In our setting $p$ is reasonably large, though it's much smaller than $n$. Let $\beta = \frac{e^p}{n}$, we have $\max_{j=1,2\ldots n} \|e_j^\top (\mathbf{HDV})\| \leq 4\sqrt{\frac{p}{n}}$ holds with failure probability at most $\frac{n}{e^p}$. In particular, when $\log(n) \ll p$ the failure probability is almost $0$.*

Next lemma is Lemma 3.4 in [1] the proof of which comes from the matrix Chernoff bound in [2].

**Lemma 3.** *(Spectral Bounds for Row Sampling). Let $\mathbf{W}$ be an $n \times p$ matrix with orthonormal columns. Define $\mathbf{M} = n \cdot \max_{j=1,2\ldots n} \|e_j^T W\|^2$. Draw $n_{subs}$ rows from $\mathbf{W}$ without replacement. Let $\mathbf{R} \in n_{subs} \times n$ be the matrix corresponding to subsampled rows. Then the smallest and largest*

*spectral value of the subsampled matrix* $\mathbf{RW}$ *are bounded by*

$$\sqrt{\frac{(1-\delta)l}{n}} \leq \sigma_p(\mathbf{RW}) \tag{3}$$

$$\sqrt{\frac{(1+\eta)l}{n}} \geq \sigma_1(\mathbf{RW}) \tag{4}$$

*with failure probability at most*

$$p \cdot \left(\frac{e^{-\delta}}{(1-\delta)^{1-\delta}}\right)^{n_{subs}/\mathbf{M}} + p \cdot \left(\frac{e^{\eta}}{(1+\eta)^{1+\eta}}\right)^{n_{subs}/\mathbf{M}} \tag{5}$$

Lemma 3 can be simplified a lot for our purpose.

**Corollary 1.** *Let* $\mathbf{W}$ *be an* $n \times p$ *matrix with orthonormal columns. Define* $\mathbf{M} = n \cdot \max_{j=1,2...n} \|e_j^\top \mathbf{W}\|^2$. *Draw* $n_{subs}$ *rows from* $\mathbf{W}$ *without replacement. Let* $\mathbf{R} \in n_{subs} \times n$ *be the matrix corresponding to the subsampled rows. Then the spectral values of the subsampled matrix* $\mathbf{RW}$ *are bounded by*

$$\sqrt{\frac{(1-\delta)l}{n}} \leq \sigma_p(\mathbf{RW}) \tag{6}$$

$$\sqrt{\frac{(1+\delta)l}{n}} \geq \sigma_1(\mathbf{RW}) \tag{7}$$

*with failure probability at most*

$$2p \cdot e^{\frac{-c\delta^2 n_{subs}}{\mathbf{M}}} \tag{8}$$

*for some fixed positive constant c.*

*Proof.* By the Taylor's expansion of $\log(1-\delta)$ and $\log(1+\delta)$

$$\log\left(\frac{e^{-\delta}}{(1-\delta)^{1-\delta}}\right) = -\delta - (1-\delta)\log(1-\delta) \leq -\delta^2$$

$$\log\left(\frac{e^{\delta}}{(1+\delta)^{1+\delta}}\right) = \delta - (1+\delta)\log(1+\delta) \leq -\delta^2/4$$

replace the $\frac{e^{-\delta}}{(1-\delta)^{1-\delta}}$ and $\frac{e^{\eta}}{(1+\eta)^{1+\eta}}$ term in lemma 2 with $e^{-c\delta^2}$ and $e^{-c\eta^2}$. Set $\eta = \delta$ completes the proof. $\square$

Now we can prove Lemma 1:

*Proof.* $\Theta = \mathbf{RHD}$. Let $\mathbf{W} = \mathbf{HDX}$, note that the columns of $\mathbf{X}/\sqrt{n}$ are orthonormal. Remark 2 shows

$$\max_{j=1,2...n} \|e_j^\top \mathbf{W}/\sqrt{n}\| \leq 4\sqrt{\frac{p}{n}} \tag{9}$$

holds with failure probability $\frac{n}{e^p}$. Let $\mathbf{M} = 16p = n \cdot \max_{j=1,2...n} \|e_j^\top \mathbf{W}/\sqrt{n}\|^2$. Assume equation 9 holds, Corollary 1 implies the spectral norm of $\Theta \mathbf{X}/\sqrt{n} = \mathbf{RW}/\sqrt{n}$ can be bounded by

$$\sqrt{\frac{(1-\varepsilon)n_{subs}}{n}} \leq \sigma_p(\Theta \mathbf{X}/\sqrt{n}) \tag{10}$$

$$\sqrt{\frac{(1+\varepsilon)n_{subs}}{n}} \geq \sigma_1(\Theta \mathbf{X}/\sqrt{n}) \tag{11}$$

with failure probability at most $\delta$ where $\varepsilon = \sqrt{\frac{c\log(\frac{2p}{\delta})p}{n_{subs}}}$. Equations 10, 11 implies that the singular values of the symmetric matrix $\frac{(\Theta \mathbf{X})^\top \Theta \mathbf{X}}{n}$ lie between $[\frac{(1-\varepsilon)n_{subs}}{n}, \frac{(1+\varepsilon)n_{subs}}{n}]$, or in other words,

the singular values of the symmetric matrix $\frac{(\Theta \mathbf{X})^\top \Theta \mathbf{X}}{n_{subs}}$ lies between $[1 - \varepsilon, 1 + \varepsilon]$. Noticing that $\mathbf{X}^\top \mathbf{X}/n$ is a $p \times p$ identity matrix, so Equations 10, 11 directly imply Equation 1. Finally let's compute the failure probability, i.e. the probability that the Equations 10, 11 don't hold. By Lemma 1,

$$P(\text{Equation 9 fails}) \leq \frac{n}{e^p} \tag{12}$$

By corollary 1,

$$P(\text{One of Equations 10, 11 fail}|\text{Equation 9 holds}) \leq \delta \tag{13}$$

which directly implies

$$P(\text{One of Equations 10, 11 fail and Equation 9 holds}) \leq \delta \tag{14}$$

Equations 12, 14 imply

$$
\begin{aligned}
P(\text{One of Equations 10, 11 fail}) \quad \leq \quad & P(\text{One of Equations 10, 11 fail and Equation 9 holds}) \\
& + P(\text{Equation 9 fails }) \\
\leq \quad & \frac{n}{e^p} + \delta
\end{aligned}
$$

$\square$