[Reviews · NeurIPS 2013]

Submitted by Assigned_Reviewer_5

The authors propose a method (SHRT) to accelerate Ridge Regression when the
number of features p is much greater than the number of samples n (p >> n).

The main idea is to reduce the dimensionality of the design input matrix X to
make the computation of XX^T faster, from O(n^2) to O(nplogn). This is done via
Walsh-Hadamard transforms, which can be computed in log-time.

The research idea is of high-quality and great use. The paper is well written
and technically correct. However, my main concern is the significant overlap to
the recently published paper:

"Fastfood – Computing Hilbert Space Expansions in loglinear time,
Quoc Le; Tamas Sarlos; Alexander Smola", ICML 2013.

Both submissions are very close in time, so I believe they were developed
independently. Fastfood seems completely analogous to the here proposed
SHRT algorithm, but more general:

SHRT(X) := RHDX
FASTFOOD(X) := exp(SHGPHBX)

1) The basis matrix R in SRHT is the matrix P in Fastfood.
2) The Walsh-Hadamard matrix H in SRHT is the matrix H in Fastfood.
3) The Rademacher matrix D is the matrix B in Fastfood.
4) The main difference is the setup of p >> n for SHRT. In Fastfood,
the number of random features p_subs to generate can be greater than p,
so multiple "SHRT blocks" can be built.
5) In SHRT there exists no matrix "G", which corresponds to spectral samples
of a Gaussian kernel in Fastfood. This is because in SHRT a huge feature
space is assumed, so linear kernel is used (that is, Fastfood's G and S
matrices are diagonal in SHRT). This is also the reason why SHRT
is not exponentiated.

The theoretical developments in Lemma 2 and Theorem 1 are mathematically well
executed and are novel material. This is also the case for the application of
the method to the PCA algorithm. Therefore, there is enough novelty in the
submission.

The authors should cite Rahimi & Brecht works (2007, 2008) and the Fastfood
paper.

I believe that the admission of this work demands a better connection with
existing work and (ideally) a comparison to other randomized methods.

I have read the Author Feedback.
Summary: A method to accelerate the computation of XX^T in ridge regression from O(n^2p)
to O(nplog(n)). The idea is to reduce the dimensionality of X via subsampled
randomized Hadamard tranforms. The paper exhibits some overlap with the ICML2013
work "Fastfood – Computing Hilbert Space Expansions in loglinear time", but novel
theoretical developments.

Submitted by Assigned_Reviewer_6

This paper proposes a fast algorithm for ridge regression by applying subsampled randomized Hadamard transform (SRHT), which can reduce the dimensionality and reduce the computational cost significantly. The basic idea of this algorithm is first applying the randomized Hadamard transform, and then sampling features uniformly. Also the theoretical results on the risk of the proposed algorithm is analyzed. Based on the proved theoretical results, the order of subsampled features p_subs is estimated. And the comparison between the proposed algorithm and the principal component regression is also discussed. The empirical studies on both synthetic and UCI data sets are performed to demonstrate the good performance of the proposed algorithm.
The paper presents an interesting algorithm and is easy to read. The idea of the proposed algorithm is fairly simple. The key is the efficient implementation of the subsampled randomized Hadamard transform. The idea is simple but interesting to me. The disadvantage of other random projection algorithms is the cost to generate a random projection matrix and compute the product of the projection matrix and the design matrix. The key of this paper is to integrate these two steps into one step which is implemented efficiently.
The empirical studies are a little weak. For the UCI data sets, the classification accuracy is reported. As this paper focuses on ridge regression and the theoretical bound of risk is proved, it is more reasonable to investigate the error in regression instead of classification.
A typo: in Eq. (3), the right hand side of the inequality is sqrt{c*log(2k/delta)*k)/p_subs}, while in the following discussions there no coefficient 2. I have not checked the cited references, but it seems a typo.
Summary: A simple and efficient implementation of ridge regression using the subsampled randomized Hadamard transform is proposed. The idea is simple but interesting to me; the theoretical results and the empirical results are a little weak.
Author Feedback

Author rebuttal: We would like to thank all the reviewers for the detailed and insightful reviews:


Assigned_Reviewer_4:

We would clean up the citations and correct the typos in the text! Thanks for pointing them out!

We would also discuss connection with http://arxiv.org/pdf/1109.5981v2.pdf. We were not aware of this paper.


Assigned_Reviewer_5:

We would add citation to Rahimi & Recht (2007, 2008). We only got to know about the "FastFood" paper at ICML this year, which took place after NIPS submission deadline. We would definitely add a citation to it!

"Fastfood – Computing Hilbert Space Expansions in loglinear time,
Quoc Le; Tamas Sarlos; Alexander Smola", ICML 2013.

Assigned_Reviewer_6:

Thanks for pointing out the typo; we would correct it!